

# Maternal effects, reciprocal differences and combining ability study for yield and its component traits in maize (*Zea mays* L.) through modified diallel analysis

Bonipas Antony John[1], Rajashekhar Mahantaswami Kachapur[2], Gopalakrishna Naidu[3], Sidramappa Channappa Talekar[2], Zerka Rashid[4], Bindiganavile S. Vivek[5], Nagesh Patne[6], Shiddappa Ramappa Salakinkop[7] and Prema GU[8]

[1] Genetics and Plant Breeding, Tamil Nadu Agricultural University, Coimbatore, Tamil Nadu, India
[2] Genetics and Plant breeding, AICRP on Maize, MARS, Dharwad, University of Agricultural Sciences, Dharwad, Dharwad, Karnataka, India
[3] AICRP on Soybean, MARS, Dharwad, University of Agricultural Sciences, Dharwad, Dharwad, Karnataka, India
[4] Plant Pathology, CIMMYT, Hyderabad, Telangana, India
[5] Regional Maize Breeding Coordinator, ASIA, CIMMYT, Hyderabad, Telangana, India
[6] Plant Breeding, CIMMYT, Hyderabad, Telangana, India
[7] Agronomy, University of Agricultural Sciences, Dharwad, Dharwad, Karnataka, India
[8] Plant Pathology, University of Agricultural Sciences, Dharwad, Dharwad, Karnataka, India

Corresponding authors
Rajashekhar Mahantaswami Kachapur, kachapurr@uasd.in
Bindiganavile S. Vivek, B.Vivek@cgiar.org

## ABSTRACT

Combining ability status of the inbred lines is crucial information for hybrid breeding program. Diallel or line × tester mating designs are frequently used to evaluate the combining ability. In the current study a modified diallel model was used, wherein the Griffing's combining ability effects were further partitioned to understand the effects due to maternal and reciprocal. To do this, eight parental lines of maize were crossed in full diallel method and the generated hybrids along with parents were phenotyped. The field data on the quantitative traits was analyzed using both Griffing's and the modified model to determine how well the parents' and the $F_1$ hybrids combined. For each of the traits, a sizable reciprocal and maternal variance was observed. The number of kernel rows per cob variable had a ratio of additive variance to dominance variance greater than one. All other traits including grain yield had a ratio close to zero, suggesting that non-additive gene action was primarily responsible for the genetic control of most of the traits. The narrow sense heritability was low to moderate for majority of the variables, except for number of kernel rows per cob. With the help of the improved model, it was possible to choose superior parents and cross-parent pairings with accuracy. Based on the modified general combining ability effects and maternal effects, the parental line P5 was recognized as a potential female parent and P7 as a good male parent for grain yield and yield-attributing characteristics. The cross combination of P8×P1 had the highest specific combining ability effect on grain yield. P5×P6 cross had the highest reciprocal effect. The correlation analysis implies that the Griffing's general combining ability effects and specific combining ability effects were found to be less efficient in predicting $F_1$ performance as compared to the modified model.

# INTRODUCTION

Among the cereal grains, maize (*Zea mays* L.) is most widely cultivated and adapted crop, and used for a variety of purposes, including human nutrition, poultry and animal feed, apart from a number of industrial uses (*Gupta, Hossain & Muthuswamy, 2015*). In order to meet the growing demand for maize, India needs to double its maize production by the year 2050 (*Mehta et al., 2021*). To address this challenge of increasing production maize breeders are in the quest of developing hybrids with more productive traits with desirable agronomic and physiological characteristics (*Andorf et al., 2019*). In-depth research is being done to create superior hybrids with high yielding abilities, and resistance to biotic and abiotic stresses in order to accomplish this. The productivity of hybrids depends on the genetic performance of their parents. The most difficult aspect of hybrid breeding is screening and evaluating the inbred lines, crossing and identifying the best cross combinations that result in productive hybrids (*Patil, Kachapur & Nair, 2021*). Based on test cross performance the line's adaptability, stability and combining ability can be known and this would assist in identifying suitable parents (*Bertan, Fide & Oliveira, 2007*). To get a best hybrid combination combining ability information of the parental lines and hybrid combination is very crucial (*Fasahat et al., 2016*). The performance of cross combinations is the specific combining ability (SCA), whereas the general combining ability (GCA) is defined as the average performance of a line in a series of crosses (*Sprague & Tatum, 1942*). The relative importance of advantageous alleles within a line is determined by the GCA effect, which is a measurement of a line's breeding value. The difference in allele frequencies between the parental lines in a cross combination is indicated by the SCA effect (*Zhang et al., 2012*). The ability of the parents to combine, the presence of advantageous alleles and the genetic separation between the parents all influence the performance of hybrids (*Acquaah, 2012*). Another factor to consider is the reciprocal effect (*Jumbo & Carena, 2008*), which may result from the cytoplasm's constituents or from interactions between cytoplasmic and nuclear genes. Certain lines produce superior combinations, irrespective of whether they are used as male or female, this type of interaction would vary in different materials and is known as reciprocal difference (*Fleming, Kozelnicky & Browne, 1960*). Regarding its relative significance and practical exploitation in hybrid development, there is not much efforts made. These effects do not exhibit a uniform sign between two germplasm groups and are not consistently observed across environments (*Gonzalo et al., 2007*; *Mukanga, Derera & Tongoona, 2010*).

Several maize breeding studies did not include reciprocal crosses because of lack of funding and understanding of reciprocal differences and their use. The endosperm regulates maize grain yield, so understanding reciprocal effects (RECs) is crucial, thus these effects were thoroughly researched in maize ever since hybrids were created (*Fleming, Kozelnicky & Browne, 1960*; *Pollmer, Klein & Dhillon, 1979* and *Santos et al., 2017*). Additionally, it has been shown that estimates of SCA and GCA effects are influenced by the maternal and

reciprocal effects (*Mahgoub, 2011* and *Yao et al., 2013*). In addition, a better comprehension of maternal effects would also enable a better selection response (*Falconer, 1996*).

Griffing suggested use of diallel method to estimate the combined effects of combining ability and genetic variance. Griffing's method 1 and 3 has the distinction of allowing estimation of the reciprocal and maternal effects. However, this method (1 and 3) assumes that they are likely to be similar, as suggested by *Yates (1947)* and estimates the general and specific combining ability effects based on their average effect of parents when used as female or male parents or in cross combinations. Regardless of whether a parent is used as a female or male parent, the fixed model of diallel analysis estimates one GCA effect for each parent and one SCA effect for each cross combination. The contributions of each parent are included in this estimate as a whole. The modified model proposed by *Mahgoub (2011)* can give more accurate estimates of GCA and SCA as well as information on maternal and RECs (*Ghareeb & Fares, 2016*) by partitioning the combining ability effects. It also reveals how the estimation of the GCA and SCA effects is impacted by the inclusion of reciprocal crosses in the diallel. As a result, the current study's main objective is to (i) compare the effects of GCA and SCA before and after partitioning, (ii) estimate the maternal and reciprocal effects and their relationship to GCA and SCA, and (iii) to understand the relative contributions of female and male parents in the cross combinations and to identify promising female and male parents.

## MATERIALS AND METHODS

### Field evaluation of diallel crosses

The selected eight parental inbred lines of maize (Table 1) were planted during October, 2019 (Rabi season) and crossing was done in full diallel method (including reciprocal crosses) and harvested in the month of April, 2020. The harvested ears were shelled, processed and the seed packets were prepared. These, 56 $F_1$ hybrids along with four checks: 900-M Gold, NK-6240, GH-0727 and GH-150125 and the parent lines were planted in the month of June, 2020 (Kharif season) at the All India Co-ordinated Maize Improvement Project, UAS, Dharwad in completely randomized block design with three replications for evaluation. The specific weather conditions for the growing season are provided in Table 2. The soil at the experimental site was medium-deep black soil (Vertic Inseptisol) and evaluation was under optimal situation. All the genotypes (hybrids and parental lines) were raised in two rows of four meters length by following a spacing of 60 cm × 20 cm. The recommended maize package of practices for maize were followed to raise a healthy crop (*Zaidi et al., 2017*; http://aicrpmaize.icar.gov.in/karnataka;k0162_3.pdf (icar.gov.in)).

### Data collection

The data on eight quantitative traits was measured across all the replications. The traits studied were: days to 50% tasseling (DTT), days to 50% silking (DTS), number of kernel rows per cob (NKRC), number of kernels per row (NKR), cob girth (CG) (cm), cob length (CL) (cm), hundred grain weight (HGW) (g) and grain yield (GY) (q/ha). Below is a detailed description of the method used to measure these traits.

**Table 1  Parental lines of maize used their pedigree details and the cross combinations evaluated.**

| Parental lines | Pedigree details | Parents | Cross combinations[*] | | | | | | | |
|---|---|---|---|---|---|---|---|---|---|---|
| | | | P1 | P2 | P3 | P4 | P5 | P6 | P7 | P8 |
| IMIC 69 ($P_1$) | VL 175118 | P1 | – | P2XP1 | P3XP1 | P4XP1 | P5XP1 | P6XP1 | P7XP1 | P8XP1 |
| IMIC 68 ($P_2$) | VL 18797 | P2 | P1XP2 | – | P3XP2 | P4XP2 | P5XP2 | P6XP2 | P7XP2 | P8XP2 |
| CI 4 ($P_3$) | Pop27-C5-HS-29-1-1-# | P3 | P1XP3 | P2XP3 | – | P4XP3 | P5XP3 | P6XP3 | P7XP3 | P8XP3 |
| IMIC 02 ($P_4$) | VL162291 | P4 | P1XP4 | P2XP4 | P3XP4 | – | P5XP4 | P6XP4 | P7XP4 | P8XP4 |
| IMIC 40 ($P_5$) | VL 18780 | P5 | P1XP5 | P2XP5 | P3XP5 | P4XP5 | – | P6XP5 | P7XP5 | P8XP5 |
| CTLB 01 ($P_6$) | VL 18718 | P6 | P1XP6 | P2XP6 | P3XP6 | P4XP6 | P5XP6 | – | P7XP6 | P7XP6 |
| CTLB 02 ($P_7$) | VL 175029 | P7 | P1XP7 | P2XP7 | P3XP7 | P4XP7 | P5XP7 | P6XP7 | – | P8XP7 |
| CML 451 ($P_8$) | CML 451 | P8 | P1XP8 | P2XP8 | P3XP8 | P4XP8 | P5XP8 | P6XP8 | P7XP8 | – |

Notes.

[*]The cross combinations below the diagonal are straight crosses while above the diagonal are reciprocals.

Crosses represented as 1×2 in further tables means cross between P1×P2.

**Table 2  Details of the weather parameters during the crop growing season.**

| Month | Max.Temp (°C) | Min.Temp (°C) | RH Max (%) | RH Min (%) | Rainfall (mm) | Rainy days |
|---|---|---|---|---|---|---|
| June -2020 | 29.4 | 21.3 | 87.3 | 75.1 | 85.70 | 14.0 |
| July -2020 | 27.51 | 20.84 | 89.96 | 81.61 | 126.60 | 18 |
| August - 2020 | 26.14 | 20.43 | 92.51 | 87.32 | 323.60 | 28 |
| September - 2020 | 28.43 | 20.44 | 89.1 | 79.16 | 186.00 | 17 |
| October -2020 | 29.14 | 20.01 | 87.90 | 76.03 | 202.00 | 13 |
| November -2020 | 29.44 | 16.98 | 75.16 | 50.06 | 0.60 | 1 |
| Mean | 28.35 | 20.00 | 86.99 | 74.88 | – | – |
| Total | – | – | – | – | 924.5 | 91 |

DTT: Number of days from the day of sowing to the day on which 50 per cent of the plants in the treatment showed anthesis was recorded as days to 50% tasseling.

DTS: Number of days from the day of sowing to the day on which 50 per cent of the plants in the treatment showed silk emergence was recorded as days to 50% silking.

NKR: The average number of kernels per row from five cobs from the base to tip of ear counted physically and recorded.

NKRC: The number of kernel rows counted physically and recorded from five cobs and averaged.

CG: Average cob girth of five cobs measured using vernier caliper after removing the husk at the middle portion of the ear and measured in centimeters (cm).

CL: Average length of five cobs in centimeters (cm) after harvest measured from the base to the tip of the ear.

HGW: Weight of hundred grains drawn from a random sun-dried sample and measured in grams (g).

GY: Weight of the de-husked ears/plot recorded at the time of harvest and then converted to grain yield at 15 per cent moisture and expressed in quintals per hectare (q/ha).

## Statistical analysis

With the aid of the statistical software package Rstudio version 2022.07.1 (*RStudio Team, 2022*) and Microsoft Excel, the data gathered for the traits was put together and examined according to *Griffing (1956a)*. The diallel analysis with R package (*Yaseen, 2018*) was used to examine various effects and combining ability of method 1 and model I of the diallel analysis. The model used was

$$Y_{ij} = \mu + g_i + g_j + s_{ij} + r_{ij} + \frac{1}{c}\sum ke_{ij}$$

where $Y_{ij}$ is the observed measurement of parents i and j; $\mu$ is the population mean; gi and gj are the GCA effects of parent i and j respectively; $s_{ij}$ is the SCA effect of the cross between parents i and j; $r_{ij}$ is RECs and $e_{ij}$ is the random environmental effects associated with ij[th] individual. The restrictions imposed on the combining ability effects were: $\sum g_i = 0$ and $\sum s_{ij} = 0$ for each j (*Griffing, 1956b*).

According to *Singh & Chaudhary (1985)* the genetic components or variance due to GCA ($\sigma^2_{GCA}$), SCA ($\sigma^2_{SCA}$), and RCA ($\sigma^2_{RCA}$) were estimated. The ratio of GCA variance to SCA variance was also calculated, with ratios greater than unity indicating additive gene action and ratios less than unity indicating dominance genetic effect (*i.e.,* non-additive gene action) for the particular trait.

The additive and dominant variances, heritability was also calculated from $\sigma^2_{GCA}$, $\sigma^2_{SCA}$, $\sigma^2_{RCA}$ as follows,

$$\sigma^2_A = 2\sigma_{GCA}$$
$$\sigma^2_D = \sigma_{SCA}$$

$$H^2_{bs} = \frac{\sigma^2_A + \sigma^2_D}{\sigma^2_P}$$

$$h^2_{ns} = \frac{\sigma^2_A}{\sigma^2_P}$$

where r = Number of replications, $\sigma^2_{GCA}$ = Variance due to GCA, $\sigma^2_{SCA}$ = Variance due to SCA, $\sigma^2_{RCA}$ = Reciprocal variance, $\sigma^2$error = Error variance, $\sigma^2_P$ = Phenotypic variance, $\sigma^2_A$ = Additive variance, $\sigma^2_D$ = Dominance variance, $H^2_{bs}$ = Broad sense heritability, $h^2_{ns}$ = Narrow sense heritability (*Singh & Chaudhary, 1985*).

Baker's ratio was used to determine the relative importance of GCA and SCA effects for each trait (*Baker, 1978*).

$$\text{Baker's ratio} = \frac{2\sigma^2_{GCA}}{[2\sigma^2_{GCA} + \sigma^2_{SCA}]}$$

Griffing's model formula

Griffing's method of combining ability effects was estimated using the following model,

$$\hat{g}_i = \frac{1}{2n}(x_{i.} + x_{.i}) - \frac{1}{n^2}(x_{..})$$

$$\hat{s}_{ij} = \frac{1}{2}(x_{ij} + x_{ji}) - \frac{1}{2n}(x_{i.} + x_{.i} + x_{j.} + x_{.j}) + \frac{1}{n^2}(x_{..})$$

$$\hat{r}_{ij} = \frac{1}{2}(x_{ij} - x_{ji})$$

Modified model formula

For the precise estimation, the GCA effect is $\widehat{g_i}$ partitioned according to *Mahgoub (2011)* to estimate GCA effect for the parent when it is used as a female in its hybrid combination $\widehat{g_{fi}}$ and GCA effect for the same parent when it is used as a male in its hybrid combination $\widehat{g_{mi}}$ as follows:

$$\widehat{g_{fi}} = \frac{1}{n}(x_{i.}) - \frac{1}{n^2}(x_{..})$$
$$\widehat{g_{mi}} = \frac{1}{n}(x_{.i}) - \frac{1}{n^2}(x_{..})$$
$$\widehat{g_i} = \frac{1}{2}\left(\widehat{g_{fi}} + \widehat{g_{mi}}\right)$$

where, $\sum \widehat{g_{fi}} = 0, \sum \widehat{g_{mi}} = 0$ and $\sum \widehat{g_i} = 0$

$$\frac{1}{2}\left(\widehat{g_{fi}} - \widehat{g_{mi}}\right) = \frac{1}{2}\left[\frac{1}{n}(x_{i.}) - \frac{1}{n^2}(x_{..}) - \frac{1}{n}(x_{.i}) + \frac{1}{n^2}(x_{..})\right]$$
$$\frac{1}{2}\left(\widehat{g_{fi}} - \widehat{g_{mi}}\right) = \frac{1}{2}\left[\frac{1}{n}(x_{i.}) - \frac{1}{n}(x_{.i})\right] = \frac{1}{2}(x_{i.} - x_{.i})$$
$$\widehat{m_i} = \frac{1}{2}\left(\widehat{g_{fi}} - \widehat{g_{mi}}\right)$$

where $\sum \hat{m} = 0$.

The average difference between the $\widehat{g_{fi}}$ and $\widehat{g_{mi}}$ was proved to be equal to of maternal effects. It is exactly equal to maternal effect estimated according to *Cockerham (1963)* $\hat{m} = \left(\frac{x_{i.} - x_{.i}}{2n}\right)$. Thus, it proves the $\widehat{g_{fi}} - \widehat{g_{mi}}$ estimate provides the precise estimation of maternal effects.

Specific combining ability effect is partitioned to estimate SCA effect for the cross $\widehat{s_{ij}}$ and for its reciprocal $\widehat{s_{ji}}$ as follows:

$$\widehat{s_{ij}} = x_{ij} - \frac{1}{2}\left(x_{i.} + x_{.i} + x_{j.} + x_{.j}\right) + \frac{1}{n^2}(x_{..})$$
$$\widehat{s_{ji}} = x_{ji} - \frac{1}{2}\left(x_{i.} + x_{.i} + x_{j.} + x_{.j}\right) + \frac{1}{n^2}(x_{..})$$

where, $\sum \widehat{s_{ij}} + \widehat{s_{ji}} = 2 \ Griffing's \ \widehat{s_{ij}}$

Reciprocal effects were calculated from partitioned specific combining ability as follow,

$$r_{ij} = \frac{1}{2}\left(\widehat{s_{ij}} - \widehat{s_{ji}}\right) and \ r_{ji} = -r_{ij}$$

As a result, the difference between the SCA effect of a cross and its reciprocal equals the estimated reciprocal effect. Therefore, this difference provides a precise estimate of the reciprocal effect. Testing the significance differences was estimated according to Griffing's method.

where

$\widehat{g_i}$ = Griffing's GCA effect of i[th] parent,

$\widehat{g_{fi}}$ = Mean deviation of the i[th] parent as a female, averaged over a set of n males, from the grand mean,

$\widehat{g_{mi}}$ = Mean deviation of the i[th] parent as a male, averaged over a set of n females, from the grand mean,

$\widehat{m_i}$ = Maternal effect of i[th] parent,

$\widehat{s_{ij}}$ = SCA effect of the cross combination with i<sup>th</sup> female and the j<sup>th</sup> male parent,

$\widehat{s_{ji}}$ = SCA effect of the cross combination with j<sup>th</sup> female and the i<sup>th</sup> male parent,

$r_{ij}$ = reciprocal effect involving the i<sup>th</sup> female and the j<sup>th</sup> male parent,

$x_{ij}$ = The mean of cross resulting from crossing the i<sup>th</sup> female with the j<sup>th</sup> male,

$x_{ji}$ = The mean of cross resulting from crossing the j<sup>th</sup> female with the i<sup>th</sup> male,

$x_{i.}$ = The sum of i<sup>th</sup> female over all males,

$x_{.i}$ = The sum of i<sup>th</sup> male over all females,

$x_{j.}$ = The sum of j<sup>th</sup> female over all males,

$x_{.j}$ = The sum of j<sup>th</sup> male over all females,

$x_{..}$ = Grand total.

Correlation analysis

The mean values, mid-parent, and better-parent heterosis were correlated with the Griffing's GCA, SCA, and adjusted GCA and SCA effects. The heterosis values of straight and reciprocal hybrids (S3-6) and the mean performance of straight and reciprocal hybrids (S1-2) were listed in the supporting files. Using MS-Excel, the correlation analysis was carried out.

## RESULTS

### Analysis of variance for combining ability

To understand source of variability and how it is manifested in the experimental material the analysis of variance was computed (Table 3). The results of the statistical analysis of variance showed that the treatments were significant for each of the traits, which suggests that the experimental material was varied. The GCA was significant for all the examined traits, but for the traits DTT, DTS and NKRC it was higher than the SCA indicating that these traits are regulated by additive gene action. SCA was also significant for all the traits, but for NKR, CL, HGW and GY it was higher than GCA, indicating the significance of non-additive gene action in regulating these traits. For cob girth (CG) even though GCA and SCA were significant but their effects were very low. The value of the maternal effect is demonstrated by the significance of RECs in every trait. Maternal and non-maternal components make up the reciprocal effect but it is the maternal component that is important.

### Genetic parameters

The information on genetic parameters and heritability are shown in Table 4 with their estimates. The $\sigma^2e$ was significantly lower indicating that all the traits had lesser impact from the environment. The NKRC trait had a $\sigma^2A$ value higher than $\sigma^2D$, indicating that additive genes were primarily responsible for this trait. The fact that SCA variance for all the traits was greater than GCA variance shows that non-additive gene action plays an important role in governing all traits primarily (Table 4). The additive to dominance variance ratio for NKRC was greater than unity, indicating additive gene action. The trait DTS (H²bs = 94.54) showed the highest broad sense heritability. All other traits also demonstrated high broad sense heritability. The narrow sense heritability of all other traits was low, except

**Table 3** Analysis of variance for the quantitative traits studied.

| | $df$ [a] | DTT | DTS | NKR | NKRC | CG | CL | HGW | GY |
|---|---|---|---|---|---|---|---|---|---|
| Replications | 2 | 6.94 | 6.52 | 2.78 | 0.17 | 0.03 | 2.84 | 2.89 | 149.49 |
| Treatments | 63 | 21.43[**,b] | 19.58[**] | 64.03[**] | 2.52[**] | 0.42[**] | 12.99[**] | 54.05[**] | 1068.46[**] |
| GCA[a] | 7 | 79.75[**] | 51.52[**] | 51.93[**] | 11.83[**] | 0.81[**] | 5.61[**] | 55.93[**] | 590.83[**] |
| SCA[a] | 28 | 21.20[**] | 28.18[**] | 102.80[**] | 2.46[**] | 0.63[**] | 21.26[**] | 92.23[**] | 1496.90[**] |
| Reciprocal | 28 | 7.09[**] | 2.99[**] | 28.29[**] | 0.28[**] | 0.12[**] | 6.56[**] | 15.40[**] | 759.43[**] |
| Maternal | 7 | 18.55[**] | 3.87[**] | 51.32[**] | 0.27[**] | 0.32[**] | 12.25[**] | 37.11[**] | 1944.02[**] |
| Non-Maternal | 21 | 3.27 | 2.70[**] | 20.61[**] | 0.28[**] | 0.05[**] | 4.66[**] | 8.17[**] | 364.57[**] |
| Residual | 126 | 2.47 | 2.13 | 0.82 | 0.19 | 0.01 | 0.90 | 1.17 | 47.44 |
| CV[a] | – | 2.50 | 2.30 | 3.03 | 3.04 | 4.86 | 5.73 | 4.29 | 13.79 |
| S.Em.± | – | 0.91 | 0.85 | 0.95 | 0.25 | 0.06 | 0.55 | 0.62 | 4.30 |

**Notes.**
[a] GCA, general combining ability; SCA, specific combining ability; df, degrees of freedom; CV, Coefficient of variation.
[b] Asterisks (* and **) indicate significance at 0.05 and 0.01 probabilities, respectively.

**Table 4** Estimation of genetic parameters and heritability for the different quantitative traits.

| Traits | Genetic parameters | | | | | | | | | | |
|---|---|---|---|---|---|---|---|---|---|---|---|
| | $\sigma^2_{gca}$ | $\sigma^2_{sca}$ | $\sigma^2_{rca}$ | $\sigma^2_P$ | $\sigma^2_e$ | $\sigma^2_A$ | $\sigma^2_D$ | $\sigma^2_A/\sigma^2_D$ | H²(%) | h²(%) | Baker's ratio |
| **DTT** | 1.52 | 3.50 | 0.76 | 7.58 | 0.82 | 3.04 | 3.50 | 0.87 | 86.36 | 40.13 | 0.46 |
| **DTS** | 0.80 | 4.40 | 0.11 | 6.35 | 0.71 | 1.60 | 4.40 | 0.36 | 94.54 | 25.25 | 0.27 |
| **NKR** | 1.66 | 19.08 | 4.57 | 27.06 | 0.27 | 3.32 | 19.08 | 0.17 | 82.78 | 12.26 | 0.15 |
| **NKRC** | 0.21 | 0.35 | 0.13 | 0.91 | 0.06 | 0.41 | 0.35 | 1.17 | 83.51 | 45.26 | 0.54 |
| **CG** | 0.02 | 0.11 | 0.01 | 0.15 | 0.004 | 0.03 | 0.11 | 0.27 | 92.62 | 21.05 | 0.23 |
| **CL** | 0.36 | 3.96 | 0.94 | 5.63 | 0.03 | 0.72 | 3.96 | 0.18 | 83.12 | 12.82 | 0.15 |
| **HGW** | 1.72 | 17.04 | 2.37 | 22.99 | 0.38 | 3.45 | 17.04 | 0.20 | 89.14 | 15.00 | 0.17 |
| **GY** | 22.09 | 271.25 | 118.6 | 439.3 | 15.81 | 44.18 | 271.25 | 0.16 | 71.80 | 10.06 | 0.14 |

**Notes.**
$\sigma^2_P$, Phenotypic variance
$\sigma^2_e$, Variance of random error
$\sigma^2_A$, Additive variance
$\sigma^2_D$, Dominance variance
$\sigma^2_A/\sigma^2_D$, Ratio of additive variance to dominance variance
$H^2$ Broad sense heritability
$h^2$ Narrow sense heritability

for the NKRC ($h^2$ns = 45.26), which had a medium narrow sense heritability. The relative significance of GCA and SCA effects in predicting progeny performance was examined using Baker's ratio. For the traits like NKRC, the Baker's ratio was greater than 0.5, whereas it was less than 0.5 and close to zero for the traits NKR, CL, HSW, and GY.

## General combining ability

Griffing's method and modified method were used to estimate the GCA effects, and Table 5 shows Griffing's GCA effect and partitioned adjusted GCA values. The parental lines P1 ($-1.43, -0.72$) and P3 ($-2.47, -1.66$) are found to be good general combiners for earliness based on the estimates for the traits DTT and DTS, respectively, according to Griffing's

**Table 5 Griffing's GCA ($g_i$) and adjusted GCA effects after partitioning into male ($g_{mi}$) and female ($g_{fi}$) effects.**

| Trait | GCA | P$_1$ | P$_2$ | P$_3$ | P$_4$ | P$_5$ | P$_6$ | P$_7$ | P$_8$ | S.Ed$g_i$ |
|---|---|---|---|---|---|---|---|---|---|---|
| | $g_{mi}$[a] | −0.97**[b] | −0.22 | −1.31** | 0.23 | 0.69** | 0.11 | 0.73** | 0.73** | |
| DTT | $g_{fi}$[a] | −1.89** | 0.23 | −3.64** | 0.69** | 0.11 | 1.86** | 1.44** | 1.19** | 0.21 |
| | $g_i$[a] | −1.43** | 0.0 | −2.47** | 0.46** | 0.40 | 0.98** | 1.09** | 0.96** | |
| | $g_{mi}$ | −0.62** | −0.66** | −0.58** | −0.37 | 0.84** | −0.24 | 0.42* | 1.21** | |
| DTS | $g_{fi}$ | −0.83** | −0.58** | −2.74** | 0.17 | 0.13 | 1.51** | 0.96** | 1.38** | |
| | $g_i$ | −0.72** | −0.62** | −1.66** | −0.10 | 0.48** | 0.63** | 0.69** | 1.30** | |
| | $g_{mi}$ | −0.22 | −0.88** | 1.46** | −1.32** | −1.33** | 1.60** | 2.11** | −1.41** | |
| NKR | $g_{fi}$ | −1.73** | 1.31** | −0.64** | −0.64** | 1.84** | −1.17** | 1.91** | −0.88** | 0.12 |
| | $g_i$ | −0.97** | 0.21 | 0.41** | −0.98** | 0.25** | 0.21 | 2.01** | −1.15** | |
| | $g_{mi}$ | −0.87** | 0.07 | 0.08 | −0.15* | −0.02 | 0.20** | 0.80** | −0.11* | |
| NKRC | $g_{fi}$ | −0.55** | 0.10 | −0.39** | −0.04 | 0.33** | −0.15* | 1.08** | −0.40** | |
| | $g_i$ | −0.71 | 0.0 | −0.15** | −0.09 | 0.15** | 0.03 | 0.94** | −0.25** | |
| | $g_{mi}$ | 0.49** | −0.92** | 0.07 | 0.01 | −0.76** | 0.75** | 0.45** | −0.11 | |
| CL | $g_{fi}$ | −0.49** | 0.35** | −0.98** | −0.16 | 0.67** | −0.33* | 0.93** | 0.01 | |
| | $g_i$ | 0.00 | −0.28** | −0.45** | −0.07 | −0.04 | 0.21 | 0.69** | −0.05 | |
| | $g_{mi}$ | 0.03 | −0.10** | 0.04** | 0.04** | 0.10 | 0.04** | 0.05** | −0.20** | |
| CG | $g_{fi}$ | −0.17** | 0.06** | −0.11** | 0.02 | 0.36** | −0.10** | 0.17** | −0.23** | |
| | $g_i$ | −0.07** | −0.02 | −0.03 | 0.03 | 0.23** | −0.03 | 0.11** | −0.21** | |
| | $g_{mi}$ | 0.39* | −0.92** | −0.11 | 1.70** | 0.70** | −0.61** | −1.11** | −0.05 | |
| HGW | $g_{fi}$ | −2.80** | 0.83** | −1.11** | 0.70** | 3.02** | −0.86** | −0.61** | 0.83** | |
| | $g_i$ | −1.20** | −0.05 | −0.61** | 1.20** | 1.86** | −0.73** | −0.86** | 0.39** | |
| | $g_{mi}$ | −1.08 | 1.65 | 5.10** | 6.55** | −1220** | 7.21** | 3.42** | −1065** | |
| GY | $g_{fi}$ | −2.70** | −4.11** | 0.21 | −3.04** | 13.90** | −7.97** | 6.54** | −2.81** | |
| | $g_i$ | −1.89 | −1.23 | 2.65** | 1.75 | 0.85 | −0.38 | 4.98** | −6.73** | |

**Notes.**

[a] $g_i$-GCA effect estimated following Griffing's procedure, $g_{mi}$–Adjusted GCA after partition when parent used as male, $g_{fi}$- Adjusted GCA after partition when parent used as female.

[b] Asterisks (* and **) indicate significance at 0.05 and 0.01 probabilities, respectively.

GCA effects ($g_i$). For grain yield the parental line P7 recorded significantly highest and positive GCA effects for GY (4.98). Apart from this, the line P7 also recorded significant GCA values for other yield contributing traits like NKR (2.01), NKRC (0.94), CL (0.69), and CG (0.11). This was followed by the parental lines P3 (2.65) and P4 (1.75) recording significant values for GY. Therefore, these three lines P7, P3 and P4 can be used as source of good general combiners for grain yield trait. Similarly, the parental lines P5 (1.86) and P4 (1.2) recorded highest and significant GCA values for HGW and were found to be good general combiners.

How a particular line will behave as a male and female parent in the hybrid combination can be determined by comparing the adjusted GCA values after partitioning into male ($g_{mi}$) and female GCA ($g_{fi}$) effects (Table 5). The parental line P7 exhibited a significant GCA effect for GY [$g_{mi}$ (3.42) and $g_{fi}$ (6.54)] followed by for other traits also NKR [$g_{mi}$ (2.11) and $g_{fi}$ (1.91)], NKRC [$g_{mi}$ (0.80) and $g_{fi}$ (1.08)], CG [$g_{mi}$ (0.05) and $g_{fi}$ (0.17)] and CL[$g_{mi}$ (0.45) and $g_{fi}$ (0.93)] respectively based on the adjusted values. The parental line

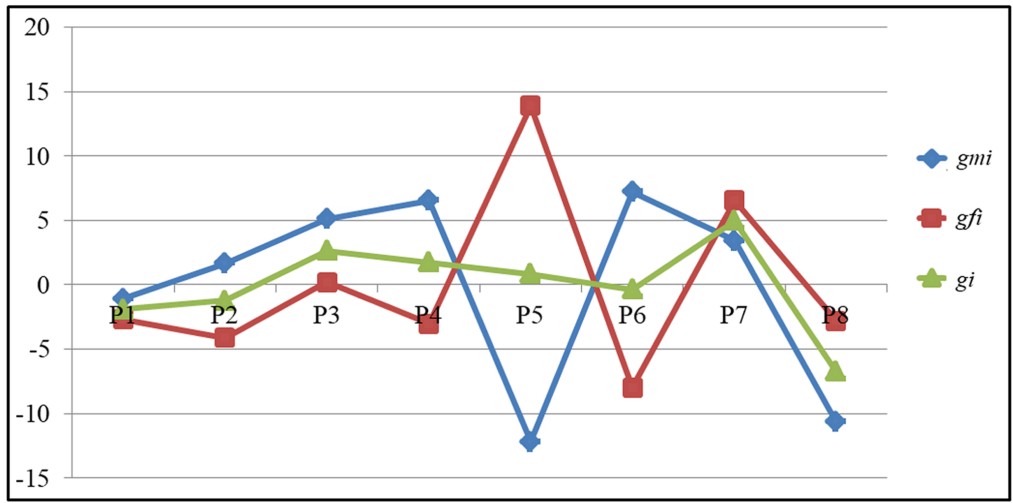

**Figure 1** The comparison of *GCA* values among parents for grain yield.

P5 recorded highest and significant breeding value in a negative direction [$g_{mi}$ (−12.20)] when used as male parent for GY. It recorded highest and significant values in a positive direction [gfi (13.90)] when used as a female parent, as opposed to gi (0.85), indicating that it is a potential line when used as a female parent for grain yield as opposed to other lines. Similarly, among the remaining parental lines P6 (7.21), P4 (6.55), P3 (5.10) followed by P7 (3.42) recorded significant $g_{mi}$ values for GY in a positive direction as compared to $g_i$ (−0.38, 1.75, 2.65, and 4.98). Likewise, for HGW the parental lines P5 (3.02), P2 (0.83), and P8 (0.83) recorded high GCA values when used as females as compared to $g_{mi}$ (−0.92, 0.70, and −0.05). Only one parental line P4 recorded highest GCA for the HGW trait when used as a male parent (1.70), indicating that it contributes more when used as a male parent. Griffing's GCA and partitioned GCA values of all eight parents is presented in Fig. 1. The figure clearly indicates that the parental lines P5 and P6 behaved differently when used as female and male parent based on the partitioned GCA values as compared to Griffing's value for GY, while the parental lines P1 and P7 did not differ significantly for Griffing's GCA and partitioned GCA values for GY. These results highlight that different genetic background (both nucleus and cytoplasmic factors) of a line plays a crucial role in deciding the breeding value of a crop.

## Maternal effect

The adjusted maternal effects, which are shown in Table 6, were determined by averaging the $g_{fi}$ and $g_{mi}$ differences. The average of the female over all the males is typically used to estimate the maternal effect. Estimating maternal effects for some specific cross combinations might be more crucial than using the ratio of all females to all associated males as a baseline. If it estimates taking into account all males over all females, it may underestimate the maternal effect of a few cross combinations. The estimation of the reciprocal effects follows the partitioning of the maternal effects, which results from the

**Table 6 Estimation of maternal effects for different traits among the parental lines.**

| Parental lines | DTT | DTS | NKR | NKRC | CL | CG | HGW | GY |
|---|---|---|---|---|---|---|---|---|
| **P₁** | −0.46 | −0.10 | −0.76**[a] | 0.16** | −0.49** | −0.10** | −1.59** | −0.81 |
| **P₂** | 0.23 | 0.04 | 1.09** | 0.02 | 0.63** | 0.08** | 0.88** | −2.88* |
| **P₃** | −1.17** | −1.08** | −1.05** | −0.23** | −0.53** | −0.07** | −0.50* | −2.45* |
| **P₄** | 0.23 | 0.27 | 0.34* | 0.05 | −0.09 | −0.01 | −0.50* | −4.79** |
| **P₅** | −0.29 | −0.35 | 1.59** | 0.18** | 0.71** | 0.13** | 1.16** | 13.05** |
| **P₆** | 0.88** | 0.88** | −1.38** | −0.17** | −0.54** | −0.07** | −0.13 | −7.59** |
| **P₇** | 0.35 | 0.27 | −0.10 | 0.14* | 0.24 | 0.06** | 0.25 | 1.56* |
| **P₈** | 0.23 | 0.08 | 0.27 | −0.14* | 0.06 | −0.02 | 0.44* | 3.92* |
| **S.Ed. $g_i$** | 0.21 | 0.20 | 0.12 | 0.06 | 0.13 | 0.02 | 0.15 | 0.93 |

**Notes.**
[a] Asterisks (* and **) indicate significance at 0.05 and 0.01 probabilities, respectively.

estimation of the maternal effects on a hybrid combination basis rather than on the average of all the associated male parents.

The findings (Table 6) indicated that the parental line P3 recorded significant maternal effects but in an unfavorable negative direction for all of the quantitative traits measured. Apart from that none of the parental lines recorded either completely negative or positive maternal effect values for all the traits. For grain yield the line P5 recorded highest and significant maternal effects (13.05) in a positive direction followed by P8 (3.92) and P7 (1.56). The same line (P5) also recorded significant values for maternal effects in a positive direction for all other traits except DTT and DTS. On the other hand P6 recorded significant maternal effects in a negative direction for GY (−7.59) followed by P4 (−4.79). The line P6 recorded significant maternal effects but in a negative direction for yield contributing traits NKR, NKRC, CL and CG. For DTT and DTS the line recorded significant maternal effects in a positive direction. For maturity traits *i.e.,* DTT and DTS the maternal effect observed in P3 (−1.17 and −1.08) was in a negative direction, thus the line P3 may be used as a female parent while developing the early maturing hybrids.

## Specific combining ability

The partitioning of SCA provides additional information, such as the SCA of the straight cross and its reciprocal cross. Griffing's SCA estimation assumes a single SCA value for a cross combination. The hightest value of Griffing's SCA effect for grain yield was observed in the cross combination of P4×P7 (30.56) (Table 7). However, there was no significant difference between Griffing's SCA and the adjusted straight and reciprocal cross SCA values for the cross P4×P7. Many crosses revealed noticeable variations between Griffing's SCA and adjusted straight and reciprocal cross SCA values (P1×P8, P4×P5 and P2×P5). For other traits, some of the crosses did not show any variation between Griffing's SCA and adjusted straight and reciprocal cross SCA values *viz.,* for NKRC and CL (P4×P7 and P7×P8), HGW and CG (P4×P8).

The comparison of SCA for GY of top performing hybrids were depicted in the Fig. 2. The trend of Griffing's SCA and the portioned effects clearly indicates the differences of the Griffing's analysis and the partitioning methods. The cross P5×P6 recorded significant

**Table 7** Griffing's SCA effects (Griffing's *sij*) and partitioned SCA of straight (*Adjsij*) and their reciprocal crosses (*Adjsji*), the maximum and minimum values along with the best cross combination for each of the traits.

| Particulars | DTT | | | | DTS | | | | NKR | | | |
|---|---|---|---|---|---|---|---|---|---|---|---|---|
| | Cross | Griffing'ss$_{ij}$[a] | Adjs$_{ij}$[a] | Adjs$_{ji}$[a] | Cross | Griffing'ss$_{ij}$ | Adj s$_{ij}$ | Adj s$_{ji}$ | Cross | Griffing'ss$_{ij}$ | Adj s$_{ij}$ | Adj s$_{ji}$ |
| Min | 1x8 | −2.88** | −2.55** | −3.21** | 1x8 | −2.15** | −1.48** | −2.82** | 5X8 | −3.16** | −3.76** | −2.56** |
| Max | 1X3 | 1.22* | 2.22** | 0.22 | 3X8 | 0.79 | −0.55 | 2.12** | 1x8 | 5.82** | 4.47** | 7.17** |
| S.Eds$_{ij}$ | 0.16 | | | | 0.53 | | | | 0.33 | | | |

| | NKRC | | | | CL | | | | CG | | | |
|---|---|---|---|---|---|---|---|---|---|---|---|---|
| | Cross | Griffing'ss$_{ij}$ | Adj s$_{ij}$ | Adj s$_{ji}$ | Cross | Griffing'ss$_{ij}$ | Adj s$_{ij}$ | Adj s$_{ji}$ | Cross | Griffing'ss$_{ij}$ | Adj s$_{ij}$ | Adj s$_{ji}$ |
| Min | 6×7 | −0.82** | −1.02** | −0.62** | 6×7 | −1.55** | −3.67** | 0.58 | 5×8 | −0.52** | −0.48** | −0.57** |
| Max | 4×7 | 0.90** | 0.90** | 0.90** | 1×2 | 2.40** | 0.51 | 4.29** | 1×8 | 0.41** | 0.29** | 0.53** |
| S.Eds$_{ij}$ | 0.34 | | | | 0.04 | | | | 0.39 | | | |

| | HGW | | | | GY | | | |
|---|---|---|---|---|---|---|---|---|
| | Cross | Griffing'ss$_{ij}$ | Adj s$_{ij}$ | Adj s$_{ji}$ | Cross | Griffing'ss$_{ij}$ | Adj s$_{ij}$ | Adj s$_{ji}$ |
| Min | 5×8 | −4.67** | −3.67** | −5.67** | 5×8 | −12.55** | −11.38** | −13.72** |
| Max | 1×8 | 5.64** | 3.39** | 7.89** | 4×7 | 30.56** | 29.32** | 31.81** |
| S.Eds$_{ij}$ | 0.39 | | | | 2.49 | | | |

**Notes.**
Two asterisks (**) indicate significance at a 0.01 probability level.

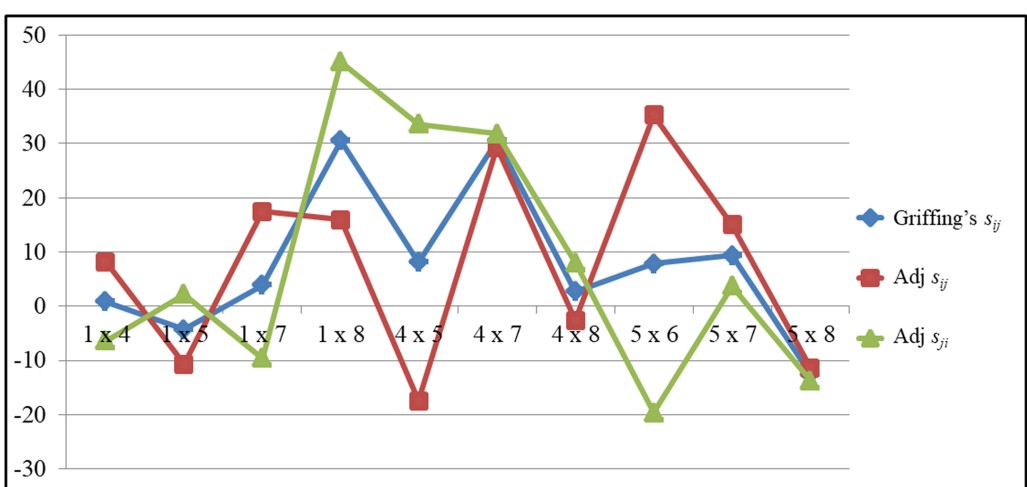

**Figure 2** The comparsion of *SCA* values among top performing hybrids for grain yield.

SCA values for GY (35.30) in a positive direction for the adjusted straight cross. But for the adjusted reciprocal crosses it recorded significant SCA values (−19.63) in a negative direction. Griffing's SCA value was significant and in a positive direction (7.83). In contrast to this in the cross P4×P5, the reciprocal cross recorded significant SCA values for GY in a positive direction (33.60) and in the straight cross it was in a negative direction (−17.45), while the Griffing's SCA effects was in a positive direction (8.07). The same trend was observed for other traits also.

## Reciprocal effects

Estimates of RECs are given by the difference between the straight and its reciprocal cross based on SCA effects. Griffing's reciprocal effect is the same, but the partitioned value ($r_{ij} = -r_{ji}$) is in two directions. The cross-combination of P5 and P6 was found to have the greatest reciprocal effect on grain yield (Table 8) in a positive direction (+27.46). Meanwhile, the cross combination between P1 and P8 had the highest reciprocal effect for majority of the yield contributing traits NKR (+5.82), HGW (+5.64) and CG (+0.41). The same cross combination also recorded significant reciprocal effects in desirable negative direction for DTT (−2.88) and DTS (−2.15) respectively. For other traits such as NKRC, the cross P4×P7 recorded significant reciprocal effect in a positive direction (+ 0.90) and for CL the cross between P1×P2 recorded highest value +2.40.

## Correlation between heterosis, mean performance and combining ability

Mean performance, mid-parent heterosis, and better-parent heterosis (Table 9) were found to be strongly correlated with Griffing's SCA effect and adjusted SCA effect after partitioning. It is therefore appropriate to identify potential hybrids based on the adjusted SCA effects after partition, which had the highest correlation with hybrid performance and heterosis. The sum of the adjusted GCA effects ($g_{fi}$ and $g_{mi}$) and mean performance are strongly correlated. Griffing's GCA effect ($g_i$) was not consistently correlated with the phenotypic performance of hybrids, mid-parent heterosis, and better parent heterosis. In light of this, the adjusted GCA effect sum is more accurate at predicting hybrid performance than the sum after dividing the GCA into male and female GCA.

# DISCUSSION

The fundamental concepts in plant breeding, GCA and SCA, proposed by *Sprague & Tatum (1942)* have an impact on inbred line selection, hybrid breeding programs and population development. Along with combining ability, the maternal and its RECs are crucial for the choice of inbred lines as female or male parents in hybrid development. According to the reports, reciprocal differences between maize grain yield and other quantitative traits have been recorded (*Fan et al., 2014*; *Fan et al., 2018*; *Dosho et al., 2021*). Additionally, it has been noted that the estimation of both GCA and SCA effects is impacted by the presence of maternal and RECs (*Yao et al., 2013*).

The results of the analysis of variance for combining ability showed that SCA variance was greater than GCA variance, indicating that these characters have non-additive gene action. Numerous researchers (*Khan & Dubey (2015)*, *Yerva et al. (2016)* and *Bharat et al. (2020)* have also reported similar results. However, for NKRC, additive variance was higher than dominance variance, indicating that this trait is controlled by additive gene action. The importance of heterosis breeding in maize crop improvement is demonstrated by the predominance of non-additive gene action for grain yield and other yield contributing characters.

The broad-sense heritability was high for all the quantitative characters studied. In contrast, narrow-sense heritability was low for all other traits including grain yield, while it

**Table 8 The reciprocal effects among the crosses for different traits.**

| Reciprocal effects($r_{ij}$)[a] | DTT | DTS | NKR | NKRC | CL | CG | HGW | GY |
|---|---|---|---|---|---|---|---|---|
| 1x2 | 0.24 | −0.07 | 4.00** [b] | 0.18 | 2.40** | 0.19** | 2.83** | −13.16** |
| 1×3 | 1.22 | −0.36 | 2.61** | −0.18 | 1.19** | 0.06 | 4.64** | −2.32 |
| 1×4 | 0.12 | −0.42 | 0.45 | 0.25 | 0.37 | 0.08 | 2.08** | 7.29* |
| 1×5 | −1.32* | −0.84 | −0.13 | −0.19 | −1.22** | −0.09 | −1.33** | −6.47* |
| 1×6 | −0.23 | −0.98 | 0.90* | 0.23 | 0.50 | −0.11* | −0.48 | 9.15** |
| 1×7 | −0.01 | −0.38 | −1.74** | 0.52** | −0.91 | 0.10* | 0.64 | 13.57** |
| 1×8 | −2.88** | −2.15** | 5.82** | 0.62** | 2.00** | 0.41** | 5.64** | −14.56** |
| 2×3 | −0.05 | −0.63 | 0.72 | 0.12 | 0.07 | 0.10* | 0.48 | −10.25** |
| 2×4 | −0.48 | −0.69 | 1.81** | 0.36* | 0.72 | 0.08 | −0.33 | 5.83* |
| 2×5 | −0.92 | −0.78 | 3.93** | 0.77** | 1.34** | 0.27** | 2.02** | −21.91** |
| 2×6 | −0.51 | −0.59 | 3.07** | −0.16 | 0.50 | 0.07 | 1.11** | −1.04 |
| 2×7 | −1.11 | −0.65 | 1.02** | 0.13 | 0.69 | 0.04 | 0.98* | −7.65** |
| 2×8 | −0.48 | −0.59 | −1.27** | −0.38* | −0.97* | 0.00 | −1.27** | −1.19 |
| 3×4 | −0.34 | −0.82 | −0.29 | −0.40* | −0.89* | 0.03 | −0.52 | 0.00 |
| 3×5 | 0.06 | 0.10 | −0.15 | 0.09 | 0.19 | 0.13** | 2.08** | −16.15** |
| 3×6 | −0.19 | −0.21 | −0.88 | 0.38* | 0.13 | −0.08 | −1.83** | 3.67 |
| 3×7 | −0.13 | −0.28 | 0.53 | −0.63** | 0.14 | −0.13** | 0.05 | −10.52** |
| 3×8 | 0.16 | 0.79 | 2.23** | 0.56** | 0.73 | 0.28** | 2.30** | −9.14** |
| 4×5 | 0.12 | 0.37 | −0.38 | −0.31 | 0.59 | 0.26** | 3.02** | −25.53** |
| 4×6 | −0.80 | −0.61 | 1.06** | −0.28 | 1.14** | −0.01 | 1.11** | 6.88* |
| 4×7 | −1.40* | −2.01** | 3.62** | 0.90** | 2.22** | 0.19** | 1.98** | −1.24 |
| 4×8 | −1.28* | −1.44* | 3.54** | 0.13 | 1.87** | 0.36** | 4.48** | −5.34 |
| 5×6 | −1.90** | −1.86** | −0.32 | 0.87** | 0.53 | 0.16** | 0.70 | 27.46** |
| 5×7 | −2.01** | −1.76** | 1.03** | −0.24 | 0.58 | 0.03 | 0.33 | 5.67* |
| 5×8 | 0.79 | −0.69 | −3.16** | −0.34 | −1.18** | −0.52** | −4.67** | 1.17 |
| 6×7 | 0.08 | 0.27 | −1.23** | −0.82** | −1.55** | −0.11* | −1.58** | −9.71** |
| 6×8 | −1.13 | −0.84 | 1.78** | 0.08 | 0.69 | 0.21** | 1.92** | −4.88 |
| 7×8 | −0.57 | −0.57 | 3.43** | 0.87** | 2.03** | 0.34** | 1.80** | 2.60 |
| S.Ed$r_{ij}$ | 0.64 | 0.36 | 0.37 | 0.03 | 0.39 | 0.05 | 0.44 | 2.81 |

**Notes.**

[a]There are $r_{ij}$ values, whereas $r_{ji}$ had the same values with different sign ($r_{ij} = -r_{ji}$).

[b]Asterisks (* and **) indicate significance at 0.05 and 0.01 probabilities, respectively.

was moderate for NKRC. The value of Baker's ratio provides the overall view of the inbred lines used in the hybrid development in terms of their combining ability. Baker's ratio for grain yield and other yield-attributing traits was less than 0.5 and almost zero, indicating that SCA was a more reliable predictor of hybrid performance. A lower Baker's ratio suggests preponderance of non-additive gene action in the inbred lines expressed as SCA values majorly contributing to hybrid performance. Thus, Baker's ratio provides additional support for genetic variance estimates implying the importance of heterotic breeding in improving maize grain yield. A lower Baker's ratio value for grain yield was also noted by *Kayaga et al. (2017)*. The correlation study suggests, that the prediction based on GCA would be more accurate. The current study found significant combining ability effects and

**Table 9** Correlation analysis of combining ability effects, hybrid mean performance and heterosis.

| Correlation | DTT | DTS | NKR | NKRC | CL | CG | HGW | GY |
|---|---|---|---|---|---|---|---|---|
| Mean *versus* | | | | | | | | |
| Griffing'ss$_{ij}$[a] | 0.06 | 0.64[**b] | 0.53[**] | 0.64[**] | 0.67[**] | 0.63[**] | 0.59[**] | 0.58[**] |
| Adj$s_{ij}$[a] | 0.54[**] | 0.66[**] | 0.91[**] | 0.66[**] | 0.96[**] | 0.69[**] | 0.80[**] | 0.95[**] |
| Griffing's SGCA[a] | 0.71[**] | 0.52[**] | 0.37 | 0.52[**] | 0.33 | 0.35 | 0.57[**] | 0.34 |
| AdjSGCA[a] | 0.86[**] | 0.62[**] | 0.57[**] | 0.62[**] | 0.57[**] | 0.53[**] | 0.52[**] | 0.58[**] |
| MPH[a] *versus* | | | | | | | | |
| Griffing'ss$_{ij}$ | 0.70[**] | 0.64[**] | 0.59[**] | 0.69[**] | 0.61[**] | 0.64[**] | 0.66[**] | 0.52[**] |
| Adj$s_{ij}$ | 0.83[**] | 0.86[**] | 0.77[**] | 0.91[**] | 0.84[**] | 0.82[**] | 0.67[**] | 0.63[**] |
| Griffing's SGCA | −0.67[**] | −0.49[*] | −0.15 | −0.41[*] | −0.04 | −0.63[**] | −0.18 | −0.14 |
| Adj SGCA | −0.41[*] | −0.18 | 0.11 | −0.20 | 0.23 | −0.38[*] | −0.34 | 0.07 |
| BPH[a] *versus* | | | | | | | | |
| Griffing'ss$_{ij}$ | 0.56[**] | 0.64[**] | 0.53[**] | 0.68[**] | 0.54[**] | 0.57[**] | 0.61[**] | 0.37 |
| Adj$s_{ij}$ | 0.39 | 0.51[**] | 0.66[**] | 0.80[**] | 0.75[**] | 0.75[**] | 0.81[**] | 0.40[*] |
| Griffing's SGCA | −0.83[**] | −0.65[**] | −0.24 | −0.39[*] | −0.05 | −0.52[**] | −0.08 | −0.21 |
| Adj SGCA | −0.74[**] | −0.51[**] | −0.02 | −0.24 | 0.20 | −0.34 | −0.08 | −0.02 |

Notes.

[a]MPH –Mid-parent heterosis, BPH –Better-parent heterosis, Griffing'ss$_{ij}$_ The *SCA* effect estimated following Griffing's procedure, Adj$s_{ij}$ –The adjusted *SCA* effect of straight cross after partitioning, Griffing's S*GCA* –Sum of parental *GCA* effects estimated following Griffing's procedure, Adj S*GCA* –Sum of adjusted parental *GCA* effects after partitioning.

[b]Asterisks (* and **) indicate significance at 0.05 and 0.01 probabilities, respectively.

RECs for all the traits, both general and specific. The selection of female parents in the cross combinations is much more important in the hybrid program to produce superior hybrids, according to earlier research work by *Kumar et al. (2016)*, *Sadalla, Barznji & Kakarash (2017)*, *Onejeme, Okporie & Eze (2020)* and *Suyadi, Saptadi & Sugiharto (2021)*. They found significant reciprocal variance for the majority of maize traits. Given the importance of both the maternal and non-maternal components, it is crucial to carefully choose both male and female parents for a cross-combination.

The observed differences between straight and reciprocal crosses were used to estimate maternal effects *Grami, Stefansson & Baker (1997)* as cytoplasmic genes are responsible for maternal effects. While, the interaction between nuclear and cytoplasmic genes may help to explain non-maternal effects ((*Evans & Kermicle, 2002*). Additionally, it has been stated that non-maternal effects are caused by non-additive gene action. Whereas, maternal effects are caused by additive genetic variance (*Mukanga, Derera & Tongoona, 2010*). Because of this, the current study also suggests that all of the quantitative traits under investigation are influenced by both additive and non-additive gene action, as well as reciprocal differences. Despite this, a number of scientists, including *Fleming, Kozelnicky & Browne (1960)*, *Crane & Nyquist (1967)*, and *Bhat & Dhawan (1971)* had previously reported the cytoplasmic effect in maize quantitative trait inheritance. Therefore, choosing the right male and female parent is crucial for the development of heterotic hybrids, which could be accomplished by considering elements like maternal effects and RECs and combining them while making a choice. GCA effects were partitioned into g$_{mi}$ and g$_{fi}$, revealing which line is more effective as a female or male parent. Griffing's method overestimated the breeding values of parental

lines P5 and P7 when used as male parents for grain yield and underestimated them when used as female parents (Fig. 1). They could thus be utilized as female parents in the development of hybrids. The parental lines P3, P4, and P6 could also be used as male parents because they had high $g_{mi}$ values relative to $g_i$ and $g_{fi}$. The line P5 could be used as a female parent because it had high $g_{fi}$ value for GY apart from other yield contributing traits NKR, CL, CG and HGW (Table 5).

In contrast, the $g_{mi}$ for NKR, NKRC, CL and GY were high for P7 parent. Additionally, P5 had a significant positive maternal effect and P7 had a significant negative maternal effect for yield attributes. This suggests that the estimation of GCA is impacted by the presence of maternal effects (*Fan et al., 2014*). In the meantime, maternal effect on GCA was not observed in all the parents as seen in P6, which is a better parent as male and female in terms of breeding value. Thus, the interaction of cytoplasm and nucleus does not differ similarly for all the genetic backgrounds. These findings suggest that P5 could be used as a female parent, P7 as a male parent and P6 as both as a female and male parent in the development of hybrids. The SCA effects were overestimated by Griffing's method for the crosses P1 × P8 and P4 × P5 while, it was underestimated for their reciprocal crosses for GY (Fig. 2). The reciprocal cross P8 × P1 had the highest SCA effects among the test hybrids, according to the partitioning. It should be noted that these crosses showed extremely important RECs for GY. As can be seen in the example above, the RECs have a significant impact on the estimation of the SCA effects (*Yao et al., 2013* and *Mahgoub, 2004*). A lower selection response results from the presence of maternal and reciprocal effects *Roach & Wulff (1987)*. The majority of the crosses showed significant reciprocal differences, suggesting that cytoplasmic factors and their interactions with nuclear factors influence the traits that contribute to maize GY.

Additionally, only a few crosses exhibit reciprocal and maternal effects, indicating that the breeding material used to produce these effects may vary, *i.e.,* it may be highly genotype-specific (*Fleming, Kozelnicky & Browne, 1960*). The maternal and RECs differ based on environmental factors in addition to genotypes (*Kalsy & Sharma, 1972*). In order to choose the best base material, it is necessary to precisely estimate the combining ability, maternal and RECs across the environments. By partitioning the effects as mentioned here, this could be done. In contrast to Griffing's GCA effects, the adjusted GCA effects after partitioning had a strong correlation with mean performance. As a result, according to *Worku et al. (2008)* the sum of adjusted GCA effects may be a trustworthy predictor of mean performance. Despite this, heterosis and total GCA effects did not significantly correlate (Table 9). *Yu et al. (2020)* came to the conclusion that heterosis and sum of GCA were either negatively correlated or not correlated in Yu's study comparing combining ability with heterosis pattern in a wide variety of materials. The current study's findings, in contrast show a stronger correlation between the phenotypic performance of the hybrid and the sum of the parental GCA's. Therefore, with additional validation, the sum of the parental GCA, in particular the sum of adjusted GCA values could be used to forecast $F_1$ mean performance. The adjusted SCA effects after partitioning showed an even stronger correlation with mean performance and heterosis than Griffing's SCA effects, which already showed a significant correlation with them.

Thus, compared to Griffing's SCA effects, adjusted SCA effects more accurately predicted heterosis. Non-additive gene action was predominantly governing majority of the traits which indicates for the strong correlation between SCA effects and heterosis over the sum of parental GCA. SCA effects can therefore be used accurately to predict mid-parent and better-parent heterosis as they are more significant for heterosis than GCA effects (*Devi & Singh, 2011*; *Tian, Channa & Hu, 2017*). SCA values were less accurate and less useful in predicting hybrid performance than the sum of GCA values (*Technow, 2019*; *Liu et al., 2021*).

## CONCLUSION

Based on these results, it can be concluded that maternal and reciprocal effects have an impact on maize's quantitative traits, as well as how these effects affect combining ability estimates. However, more investigation is required to ascertain the extent to which the combining ability and maternal effects influence the traits in maize. To create hybrids with the greatest potential, it would be advantageous to estimate these effects. Griffing's diallel's effects on combining ability, maternal and RECs can be precisely estimated by partitioning their GCA and SCA effects. By taking into account the maternal and reciprocal effects on hybrid performance, suitable male and female parents can be found in order to increase heterosis.

## ACKNOWLEDGEMENTS

The All India Coordinated maize improvement project at University of Agricultural Sciences, Dharwad provided all of the logistical support for the experiment's conduct, and the authors thank CIMMYT for sharing their genetic material. The authors would also like to express their gratitude to Dr. S. S. Patil and Dr. N. G. Hanumaratti for sharing their expertise on combining ability analysis. For field research and data collection, the technical staff at AICRP-Maize is also acknowledged.

### Funding

The authors received no funding for this work.

### Competing Interests

The authors declare there are no competing interests.

### Author Contributions

- Bonipas Antony John conceived and designed the experiments, performed the experiments, analyzed the data, prepared figures and/or tables, and approved the final draft.
- Rajashekhar Mahantaswami Kachapur conceived and designed the experiments, performed the experiments, analyzed the data, authored or reviewed drafts of the article, and approved the final draft.
- Gopalakrishna Naidu analyzed the data, prepared figures and/or tables, authored or reviewed drafts of the article, and approved the final draft.
- Sidramappa Channappa Talekar analyzed the data, prepared figures and/or tables, authored or reviewed drafts of the article, and approved the final draft.
- Zerka Rashid analyzed the data, prepared figures and/or tables, authored or reviewed drafts of the article, and approved the final draft.
- Bindiganavile S. Vivek analyzed the data, prepared figures and/or tables, authored or reviewed drafts of the article, and approved the final draft.
- Nagesh Patne analyzed the data, prepared figures and/or tables, authored or reviewed drafts of the article, and approved the final draft.
- Shiddappa Ramappa Salakinkop analyzed the data, prepared figures and/or tables, authored or reviewed drafts of the article, and approved the final draft.
- Prema G.U. analyzed the data, prepared figures and/or tables, authored or reviewed drafts of the article, and approved the final draft.

## Data Availability

The raw data are available in the Supplemental Files.

## Supplemental Information

Supplemental information for this article can be found online at http://dx.doi.org/10.7717/peerj.17600#supplemental-information.

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
