# Peer review of "Maternal effects, reciprocal differences and combining ability study for yield and its component traits in maize (Zea mays L.) through modified diallel analysis"

_PeerJ, doi:10.7717/peerj.17600_

## Round 0.1 · original submission · Major Revisions

In your manuscript you use the formulas developed by Mahgoub (2011), who modified Griffing's method and compared the differences of the method as he did. The only new thing you have done here, unlike Mahgoub, is simple correlations. While your research is not new to science, it contains valuable information for parent selection in maize breeding. It is well organized and written, but needs some corrections. The reviewers have examined your manuscript thoroughly and have made all the criticisms it be should have. In addition to them, my comments can be found below.

General comments
- You should show the abbreviations as you first defined them throughout the text and in the Tables. Do not use different abbreviations. Check and correct them all.
- Use only "male and female" instead of "seed or pollen".
- Why did you not calculate the mid-parent and best parent heterosis values of reciprocal crosses? Their absence is an important shortcoming of the study.

Corrections in the text
- Line 55-57: Move the references to the end of the sentence so that they do not break the sentence. Make all of them.
- Line 79: Eight parents. Not sixteen.
- Line 80-84: You planted 8 parents in the 2019 Kharif season to make the hybrids. You harvested the hybrid combinations in the Rabi season. I don't know if the hybridization time falls in the Rabi season. This sentence is a bit confusing. I think it would be better to write that only 56 hybrid combinations and their 8 parents were grown in the 2020 Kharif season.
- Line 96: There are explanations of the abbreviations above. I don't think you need to give them again, just write them as abbreviations. The same for the others that follow.
- Line 119: Write "ability" instead of "capability".
- Line 142: "Acorrding to Baker's formula," should be deleted.
- Line 213: Replace "number of kernel rows per cob" with its abbreviation. Keep the abbreviations throughout the text.
- Line 215-216: This information is clearly visible in Table 4. I think Figure 1 is redundant and should be deleted.
- Line 217: You can add the values on heritabilities (broad and narrow sense) to Table 4. I understand your effort to visualize your manuscript, but it is unnecessary. What you find is more important than how it looks. Therefore, Figure 2 should be deleted and the values should be added in Table 4.
- Line 260-266: Is it desirable to have maternal effects? Or does it only allow us to determine whether we should use the respective parent as female or male? I am not sure that the word "desirable" is necessary with this in mind. Could it mean that we can use parents with negative results as males in breeding for the trait of interest?
- Line 300-301: You have already explained the abbreviation of GCA and SCA above. Now just write them as abbreviations.
- Line 309-315: Some of these sentences were also in the introduction. I find their presence in the discussion unnecessary. Move the ones that are not in the introduction to the appropriate places in the introduction.
- Correct Table 1 to also show cross combinations.
- Why Table 5 only has the standard error of the GCA effect values that you calculated from the original Griffing procedure. It would be more appropriate to calculate it for the ones you modified as well.
- Suppl. Tables, Table 7 and Table 8, use a "cross" instead of a "comma" when showing hybrid combinations and do not just show them as numbers. Use the hybrid codes from Table 1. For example "P1 x P2".
- To improve the presentation of all the data in Table 7, you might consider presenting it as Supp Table. In the main text you could present only the highest, lowest or most significant values in Table 7 instead of all values.
- Figure 3 and 4 should be included it in the Results section. Don't just cite it in the Discussion. Explain them in the Results as well.

·

Basic reporting

The paper is written in clear and professional english.

Tables and figures are clear.

Literature references are excellent, taking into the consideration they are new and updated (2000 and above).


The article is self-contained with relevant results to hypotheses.

Experimental design

The paper is within the research of the Peerj Aims and Scope.

Research question is well defined, relevant and meaningfull. The importance of Cytoplasm and it's genes, and their interaction with nuclear genes still intrigues plant breeding so much, as so the importance of such paper is of high value.

Experimental design-Even though the paper's basics are founded on the Griffin's statistical models from mid last century, it's update and newly refreshed statistical upgrade gives us clues of the deeper impact of the usage of one and the same maize inbred line either as male or female component.
Statistical methods desribed in the research are done with sufficient detail & information to replicate.

As for the "lack", maybe greater information could be usefull in future work by adding more genetic information of used material through heterotic origins (BSSS, ID, LSC,...).

Validity of the findings

Validity of the findings are in order with conclusions well stated, linked and updated from and to original research question.

Additional comments

Since I represent breeder from temperate season in Europe, I have little knowlegde of Rabi production season. Speaking of this, I suppose metheorogical data is given for the fact difficulties of maize production in such conditions-otherwise they (metheorological data) is not mentioned anywhere in the paper. Other than this, the paper is self-contained with relevant results to hypotheses.

·

Basic reporting

The manuscript was unambiguous in its research objectives. The chosen language was equally professional; I would use the word "strong" in scientific reporting. However, the references in the body of the manuscript were not justified, but I have suggested a change to them by restructuring them. In addition, I recommend more literature references in the Introduction section. The manuscript was organized properly, adhering to the journal`s author guide; at least a little amendment will be required. Most especially, tables should have all the abbreviated descriptions on their footnotes listed and explained properly as sighted on the same table. For the raw data, I think maybe it was an oversight by the authors that the parental lines were misleading on the assumed tags given to them, such as P9, P10, etc., and that they were missing from the reporting tables. Also, those used in the tables were missing in the raw data. For the figures, they are good. Overall, the reporting was good, the objectives were clear, and it seemed to be direct in its purpose.

Experimental design

The experiment was in line with the scope of the research area and that of the journal. It was sufficient to answer the research questions (justification and hypotheses) and can be reproduced. The introduction of a modified model to Griffing`s methods and model enhanced the scientific standard developed for these breeding programs. The mathematical expressions were clear and brief, reaching out to educate the audience. I commend the authors for their detailed methodologies.

Validity of the findings

Any work on combining ability and heterosis is crucial and has to be reported due to the scarcity of or endangered local species among the population. Parental lines, especially the landraces, should be conserved and maintained because of the inherent characteristics they possess, ranging from adaptation to resistance. Therefore, this work adds to the literature of its kind. I have suggested areas where there was either misleading data or misplaced. I would want the authors to have the review I did in Word format, because I doubt the PDF can get them what I have done. A critical lookout of any data should be given when reporting. The conclusion was well stated and precise. The idea of the whole research was covered and mentioned.

Additional comments

I hope you get the suggestions I made in Word format because they are more critical of all I have said here about the manuscript.

---

## Round 0.2 · Minor Revisions

Your manuscript needs one more minor revision from the reviewer before it is accepted.

·

Basic reporting

Here, I would suggest that the manuscript is good for publishing, though, with minor revisions. The references should be focus-based while revising the manuscript. It is better and clearer now. In the raw data entry, the parental line designations (P10, P9, etc.) should be consistent with the reported ones in the manuscript, or was there a reason for it? In addition, I attached my suggested minor corrections. All the best.

Experimental design

No comment.

Validity of the findings

No comment.

---

## Round 0.3 · Minor Revisions

You copied the reviewer's sentence verbatim while making the corrections he noted. However, the "My suggested statement reads: " section should be deleted. The reviewer's comment is Line 260, in the new revised manuscript it is Line 247. You may also have missed some comments that you need to review carefully. The old Rebutal letter is stuck in the system, please upload the one containing the new answers. It will be easy to follow. I have no additional comment. I find your other corrections sufficient, but you re-read the manuscript and correct your typos errors.

---

## Round 0.4 · accepted · Accept

Your manuscript is ready for acceptance because you have made the necessary corrections, but there is a formal error in Table 7. It does not fit on the page and some data is not visible. PeerJ Staff can contact you regarding this. From my point of view, it can also be fixed at the galley proof stage.